# Modulation of the Magnetic Hyperthermia Response Using Different Superparamagnetic Iron Oxide Nanoparticle Morphologies

**DOI:** 10.3390/nano11030627

**Published:** 2021-03-03

**Authors:** Felisa Reyes-Ortega, Ángel V. Delgado, Guillermo R. Iglesias

**Affiliations:** 1Department of Applied Physics, University of Granada, 18071 Granada, Spain; adelgado@ugr.es; 2Instituto Maimónides de Investigación Biomédica de Córdoba (IMIBIC), Hospital Universitario Reina Sofía, University of Córdoba, 14004 Córdoba, Spain

**Keywords:** hyperthermia, ILP, magnetic nanoparticles, SPION, nanocubes, nanorods, SAR

## Abstract

The use of magnetic nanoparticles in hyperthermia, that is, heating induced by alternating magnetic fields, is gaining interest as a non-invasive, free of side effects technique that can be considered as a co-adjuvant of other cancer treatments. Having sufficient control on the field characteristics, within admissible limits, the focus is presently on the magnetic material. In the present contribution, no attempt has been made of using other composition than superparamagnetic iron oxide nanoparticles (SPION), or of applying surface functionalization, which opens a wider range of choices. We have used a hydrothermal synthesis route that allows preparing SPION nanoparticles in the 40 nm size range, with spherical, cuboidal or rod-like shapes, by minor changes in the synthesis steps. The three kinds of particles (an attempt to produce star-shaped colloids yielded hematite) were demonstrated to have the magnetite (or maghemite) crystallinity. Magnetization cycles showed virtually no hysteresis and demonstrated the superparamagnetic nature of the particles, cuboidal ones displaying saturation magnetization comparable to bulk magnetite, followed by rods and spheres. The three types were used as hyperthermia agents using magnetic fields of 20 kA/m amplitude and frequency in the range 136–205 kHz. All samples demonstrated to be able to raise the solution temperature from room values to 45 °C in a mere 60 s. Not all of them performed the same way, though. Cuboidal magnetic nanoparticles (MNPs) displayed the maximum heating power (SAR or specific absorption rate), ranging in fact among the highest reported with these geometries and raw magnetite composition.

## 1. Introduction

When a suspension of magnetic nanoparticles (with certain geometrical and structural properties to be discussed below) is subjected to an alternating magnetic field of suitable strength and frequency, heat is released into the suspending medium, leading to an increase in temperature. The phenomenon is known as magnetic hyperthermia (MH) and although biomedical applications were first proposed comparatively recently [1] (previously research was focused on Joule heating by induced electric fields originated by magnetic fields in the MHz frequency range [2]), intense heating by alternating magnetic fields applied to ferromagnetic materials has been used for almost a century [3,4]. The basis of MH treatment of tumors is the fact that heating the tumor cells in a temperature up to 41–46 °C for a short period of time causes apoptosis without significantly affecting the surrounding healthy native cells. The main reasons for this finding are related to the fact that the pH of the intracellular environment is lower in the tumor cells than in the healthy cells, and this makes the former more sensitive to overheating, and in addition the poor and not well-organized vascularization of the tumor cell hinders heat dissipation as compared to the normal ones [5]. The denaturalization of some cell proteins and its effect on subsequent activation and deactivation paths has also been mentioned [6,7]. Whatever the exact mechanism, it is admitted that magnetic hyperthermia allows the local treatment in a specific site of the body reducing the need for surgery or complementing it, while keeping the secondary effects of chemotherapy to a minimum. 

As a result, MH has demonstrated good in vitro results when used against a wide variety of cancer cells, either alone [8] or, mainly, in combination with (as coadjuvant of) other therapies, including chemo- and radiotherapy [9,10,11,12,13]. Specifically, it has been demonstrated that magnetic hyperthermia improves the anticancer effects of some drugs [14], and enhances the release of encapsulated antitumor drugs [15,16,17,18,19]. Furthermore, hyperthermia may favor the immune response via several mechanisms, by enhancing the migration of immune cells toward the tumor and their toxicity against malignant cells [20]. Although no similar in vivo efficacy has been demonstrated [14], there are abundant reports regarding MH (co-)treatment of tumor xenografts in mice [11,15,21,22,23,24].

All these examples show that magnetic nanoparticles (MNPs) are potent devices that can be used as therapeutic tools in oncology, particularly in hyperthermia. In order to consider the requirements for the MNPs, which can be expected to work best as hyperthermia agents, it is necessary to recall the physical bases of hyperthermia. Heat can evolve in the presence of alternating magnetic fields by eddy currents, but this mechanism can be neglected because of the small dimensions of the MNPs applicable in the biomedical field. Second, we can think of hysteresis losses related to the area of the hysteresis cycle as w=f∮BdM, where w is the power dissipated per unit volume, *f* is the frequency of the magnetic field *B*, and *M* is the material magnetization. The field frequency must be low enough to avoid tissue heating by eddy currents and significant peripheral nerves stimulation [25]. Hence, typical values of f are around 105 Hz, but there is safety limit for the product *f*·*B* when a field is applied to humans; this limit is established in 2πfB≤3800 T/s [26], meaning that the field strength is at most of a few mT. Considering that the saturation magnetization of SPION is achieved at about 10 kOe (or 1 T in air in SI units), it is clear that the hysteresis losses can be neglected because we are under the established limit. However, if the size is such that the particles are magnetic monodomains, each one will be as a permanent magnet with magnetization parallel to the easy axis [27,28]. When a number of particles are randomly fixed in a rigid matrix, the magnetization will be negligible, and the flipping between the two orientations of the magnetic moment along the easy axis at low temperatures will only occur at high strength or high frequency of an externally applied magnetic field. An approximately square hysteresis cycle can be expected. This form of magnetic behavior is called superparamagnetism, and the model described is known as Stoner-Wohlfarth description [27,28,29]. It is important to consider these mechanisms, as superparamagnetic nanoparticles have been shown to absorb much more power at physiologically relevant magnetic fields and frequencies than multi-domain particles [30].

At finite temperature, the transition between the two orientations can be thermally produced and the coercive field tends to zero, so that the cycle shows virtually no hysteresis, and the magnetization curve looks like that of a paramagnetic material, hence the denomination of this behavior. The characteristic time needed for such spontaneous inversion of magnetization (or for reaching equilibrium after application of a step magnetic field) is called Néel relaxation time, $\tau_{N}$, exponentially growing with $\left(K_{eff}V/k_{B}T\right)$ [31,32], where $K_{eff}$ is the effective magnetic anisotropy of the particles (approximately equal to 25 kJ/m^3^ for magnetite [33,34], *V* is the particle volume, and $k_{B}T$ is the thermal energy. For instance, for a magnetite sphere 20 nm in diameter, at room temperature, $\tau_{N}≃7\;s$ [35,36]). According to the so-called linear response theory, if the field frequency is $\geq 1/\tau_{N}\approx 0.1\;\mathrm{Hz}$, the magnetization of the sample will lag behind the applied field, leading to an imaginary component, $\chi^{\prime \prime }$, of the magnetic susceptibility, which is in fact equivalent to a finite area hysteresis cycle. For high frequencies, the magnetic moment will remain in its orientation and no net magnetization will show up. According to Carrey et al. [28], this simplified approach is valid only for low field strengths and highly anisotropic particles. In practical hyperthermia applications, the particles can be free to rotate under the action of the field and viscous friction torques. The process is denominated Brownian relaxation, [31,37] and its characteristic time is $\tau_{B}=3\eta V/k_{B}T$, which, for water (viscosity *η*) amounts to about 7 μs for the magnetite particles referred to above. This opens a window of field frequencies in the hundreds of kHz range for maximizing the value of $\chi^{\prime \prime }$; note for such frequencies the Néel process is irrelevant, and only friction will be ultimately responsible for heating (a more detailed discussion can be found in Refs. [38,39]).

We can now consider the question of what conditions must MNPs fulfill to be reliable hyperthermia candidates? In order to show an effective magnetic hyperthermia response, the MNPs requirements are (in addition to being non-toxic):Superparamagnetic behavior. The particles should have zero coercive field, so that magnetization is only produced by the external field, and no aggregation takes place in quiescent conditions.Appropriate particle size. Ideally, particles should be larger than 15 nm and smaller than 100 nm. The magnetization of MNPs decreases with decreasing particle size [40]. For a given magnetic material there is an optimum size that will result in enhanced hyperthermia effects [41], but in general below 15 nm the heating absorption of the nanoparticles is limited, so it is highly desirable to determine hyperthermia efficiency of MNPs with mean size ranging above the superparamagnetic limit ≥15 nm). In contrast, those above 100 nm are of lesser interest because they display high coercivity and strong remanence, both undesired. For example, for superparamagnetic MNPs, Vreeland et al. observed that for an AC excitation field of H_0_ = 36.5 kA/m and *f* = 341 kHz, the optimum size showing maximum rate of release heat was around 22 nm, which matches the theoretical prediction of the lineal response theory [42,43]. Many authors have demonstrated that the optimal size for internalization into tumor cells, especially if these MNPs are used as theranostic agents (intracellular hyperthermia) is below 100 nm [44,45].The MNPs must be highly magnetizable. The rate of change of the magnetic energy of a system under the action of a time varying magnetic field, $\dot{H}$, is given by $-\mu M\cdot \dot{H}$, with *μ* the magnetic permeability of the medium and **M** the magnetization [46].The suspension of MNPs must be colloidally stable, with low degree of aggregation or agglomeration. It has been shown that an improvement of stability brings about a better hyperthermia performance [47,48].

Shape and crystalline anisotropies: the heating efficiency of MNPs shows dependence with the crystalline and shape anisotropy, which is optimized by developing MNPs with different morphologies [49]. Along this line, cubic shaped MNPs have been synthesized because of their higher magnetization compared to the spherical MNPs [50], due to their reduced surface anisotropy. Song et al. produced and compared the heating performance of quasi-cubical and spherical Fe_3_O_4_ nanoparticles under fields of 100 kHz and 30 kA/m. For equal concentration of Fe, the heating power was far higher for quasi-cubical particles [51]. Denfeng et al. demonstrated that rod-shaped MNPs can kill cancer cells more effectively than spherical MNPs when being exposed to AMF [52]. Another study by Nemati et al. compared deformed cube (octopods) shaped MNPs with spherical nanoparticles of similar volume and demonstrated superior heating performance of the octopods [53]. The shape of MNPs can also influence their rate of uptake and toxicity [54]. A recent study of shape effects has been reported by Mohapatra et al. [55], with very significant findings. They investigated the hyperthermia performance of spheres, octahedral, cubes, and multipods sized around 10 nm, together with 25 nm rods and 80 nm wires, all with magnetite composition. The heating power increased in the order multipods < spheres < cubes < octahedral < rods < wires, indicating a large effect of shape anisotropy on the hyperthermia effect of magnetic particles. According to the Stoner-Wohlfarth model, a contribution of minor hysteresis loops in the case of magnetic wires resulted to be determinant in this case. In fact, Iacob et al. [56] in a numerical simulation of the heating rate of spheroidal particles of different dimensions and axial ratios found that magnetic shape anisotropy can surpass the magnetocrystalline anisotropy for axial ratios as low as 1.2.

These factors must be considered when designing nanoparticles for clinical applications. Higher heating efficiency would be desirable, as it would reduce the amount of nanoparticles, field strength, and frequency required to induce significant heating. Regarding their composition and crystal structure, MNPs are mostly made from ferromagnetic iron oxides, such as magnetite (Fe_3_O_4_) or maghemite (γ-Fe_2_O_3_) for biomedical applications [32], although the former are the most interesting ones, with the additional advantage that they usually show less toxic effects in the organism [12,57].

As mentioned, the control and modulation of different magnetite nanoparticles in terms of size and morphology appears very important in order to get potent magnetic hyperthermia agents for biomedical applications. In this paper it is described an easy synthetic method that allows obtaining MNPs with different morphologies, by just varying three simple variables: reaction time, reaction temperature, and solvent. The four different morphologies obtained show a mean particle size in the range 20–60 nm, which, as detailed above, is the optimum size range for hyperthermia and tumor cell endocytosis. The magnetic hyperthermia response for each of the obtained nanoparticle morphologies has been determined and compared, using magnetic field intensities and frequencies which are safe for biomedical applications [58].

## 2. Results and Discussion

### 2.1. Morphology and Particle Size Distribution

The hydrothermal methods are increasingly used for the preparation of MNPs because of their versatility in obtaining crystalline particles with controlled size and morphology in reasonable cost and good yield [59,60]. In the present work, four different nanoparticle morphologies were obtained using this method: spheres (SP), cubes (CU), rods (RO), and stars (ST). No significant evidence of aggregation was found from TEM observations for any of the MNPs dispersed in TMAH aqueous solution (Figure 1). As observed in this figure, spherical nanoparticles can be easily transformed to cubic morphology by changing the suspension solvent from hexane to ethyl acetate. Size is not modified significantly (Figure 2). However, the cubic morphology produces significant changes in the magnetic and hyperthermia properties of the particles, as will be described below.

Reaction temperature can also affect the particle morphology. The nucleation process is thermodynamically controlled and can be altered using different temperature reactions at the beginning (when nucleation process is predominant) and at the end of the reaction. This temperature change allows the production of rod morphology, keeping the dimension of the nanoparticles between 20 and 60 nm (Figure 2); specifically, the average sizes (±SD) were: 22.2 ± 0.1 nm (diameter of SP), 24.1 ± 0.4 nm (size of CU), 27 ± 7 nm (length of rods), 3.6 ± 0.5 nm (width of rods), and 64 ± 11 nm (width of ST).

The preparation of star morphology required changing the solvent, the reaction time, and the initial molar ratios of reactants (Table 1). Benzyl ether acts as antioxidant, and it is transformed into benzaldehyde and benzoic acid in the presence of oxygen. As a result, benzyl ether provides a stronger reductive environment than 1-octanol in the reaction process. Such environment facilitates the decomposition of Fe(CO)_5_, and, as a consequence the nucleation process in benzyl ether occurs earlier and faster than in 1-octanol. The lower HDA concentration versus OA induced the preferential growth along different directions, resulting in the formation of the star nanoparticles.

### 2.2. X-ray Diffraction

The iron oxide phase of the synthesized nanoparticles was identified from the XRD pattern, as shown in Figure 3, with peak positions at (line indexes between parentheses) 30.4 (220), 35.8 (311), 37.2 (222), 43.5 (400), 53.9 (422), 57.5 (511), and 63.1 (440), which are consistent with the standard data for magnetite or maghemite (RRuff data base, ID R06111157) for samples SP, RO, and CU. Star nanoparticles (ST) were identified as hematite (α-Fe_2_O_3_). Hence these particles will be neglected as possible hyperthermia agents.

### 2.3. Electrical Surface Characterization

The measurements of the electrophoretic mobility as a function of pH were carried out as a further test of the purity of the synthesized MNPs and particularly that of their surface, by comparison with the existing literature data [61]. Results were obtained in TMAH 25%wt aqueous solutions, and they are presented in Figure 4. It can be expected that nanoparticles will acquire surface charge as a result of the ionization of the adsorbed TMAH molecules. This is a strong base and the presence of hydroxyl groups associated to its dissolution produces an increase in the negative charge of SPION [62]. Thus, the isoelectric point of the investigated MNPs was in the vicinity of pH = 5.5, lower than that of bare magnetite (pH = 6.5) [63], indicating that the adsorption of OH- leads to the requirement of more acid conditions for neutralizing the surface.

### 2.4. Magnetization Data

The magnetization cycles of the three magnetite nanoparticle samples are plotted in Figure 5. The magnetization values have been normalized to the total mass of the sample and it was observed that for all the samples the magnetization tends to saturate at 120 kA/m approximately. The low-field details included show that hysteresis is negligible, and only spheres show a measurable coercivity, below 5 kA/m. The same can be said about remanence: the particles have zero remanent magnetization, except again spheres, also below 5 kA/m. It can be said hence that the particles display superparamagnetic behavior. Interestingly, nanorods show the maximum saturation magnetization (85 emu/g, close to the value corresponding to bulk magnetite, 91 emu/g [64]). In contrast, spheres display the minimum saturation magnetization. According to the study of No et al. [50], this result can be expected, as the surface anisotropy, that is, the discrepancy between surface and bulk atomic spins is larger (about double, in fact) in spherical than in cuboidal nanoparticles. Regarding rod-shaped particles, their shape provides the required anisotropy, and Mohapatra et al. [55] also found that octahedral and cubic nanoparticles displayed larger magnetic saturations than rods. Other authors have found that the shape anisotropy prevents the particles from magnetizing in directions other than the easy axis of magnetization, and this is why they found a lower saturation magnetization in rods than even in spheres [65,66]. Additionally, it must be mentioned that the fact that the demagnetization field is lower for the small than for the large axis, additional easy magnetization axes can be created, favoring magnetization for arbitrary field directions [67].

### 2.5. Hyperthermia Response

Nanospheres, nanocuboids, and nanorods exhibited an intense hyperthermia response in the range of frequencies tested. These three magnetite morphologies (all at 10 mg/mL) increase the temperature from 25 to 45 °C in a few seconds when the magnetic field is applied (Figure 6). In general, a higher frequency of the applied magnetic field increases the specific absorption rate (SAR) of the nanoparticles. The hyperthermia response is heavily influenced by the morphology of the nanoparticles. Having the same amount of iron oxide (magnetite nanoparticle concentration) and similar particle sizes, the SAR values for cuboidal nanoparticles were far superior than those for nanospheres or nanorods (Figure 7). Considering that shape anisotropy is higher for nanorods than for nanocuboids, it could be expected to obtain higher hyperthermia response for the former. A recent simulation, where it was compared the achievable hyperthermia response using nanorods and nanocubes, showed that when the average dimensions are below 60 nm, shape anisotropy is insufficient to stabilize the particles even at short timescales. Surface anisotropy or particle–particle interactions make important contributions to Néel heating in magnetic hyperthermia applications [68]. These simulated results have been experimentally corroborated in this paper.

Both SAR and Intrinsic Loss Power (ILP) have been calculated for the three magnetite samples and different field frequencies (Figure 7). Previously, it was checked for some samples that the SAR was linearly dependent on the product *H*^2^*·f*, as expected from the linear response theory. Note how ILP is indeed roughly independent of frequency in Figure 7. As observed, nanocuboidal particles showed the highest SAR, and an ILP value of 3.0 ± 0.2 nHm^2^/kg, independently of the frequency used to carry out the measurement. The fact that *K_eff_* is maximum for this geometry [55] explains this result, considering that the size of our particles allows neglecting the possible role of hysteresis. To our knowledge, this value is the highest found at these frequencies for cubic magnetite nanoparticles, suggesting that this system is very attractive as a potent hyperthermia agent that can be applied at low, biomedically safe frequencies (less than 400 kHz). Nanospheres and nanorods showed similar SAR and ILP values at low frequencies (130–180 kHz), although at 205 kHz the spherical morphology displayed a noticeable increase in both parameters, probably due to the already mentioned lack of stabilization of the nanorods.

## 3. Materials and Methods

### 3.1. Materials

All reactants used were commercially available: hexadecylamine (HDA) 98% purity, oleic acid (OA) 99% purity, iron pentacarbonyl, dibenzyl ether 98% purity, ethyl acetate (99.5% purity), tetramethylammonium hydroxide (TMAH) solution in water (25% wt), and benzyl ether (98% purity) were purchased from Sigma-Aldrich Saint Louis, MO, USA and were used as received. Absolute ethanol and n-hexane 99% reagent grade were supplied by Scharlau, Cham Germany.

### 3.2. Synthesis of Magnetic Nanoparticles

Magnetite nanoparticles were synthesized using a previously described hydrothermal method [69], modifying different reaction variables. HDA and OA were mixed in the amounts and in the solvents indicated (Table 1). The mixtures were heated up to 55 °C and stirred for 30 min. After that, the solution was cooled to room temperature and the amount of iron pentacarbonyl (Fe(CO)_5_) also indicated in the Table 1 was added and magnetic stirring continued for 60 min. Then, the solution was transferred to a 160 mL autoclave with Teflon lining and heated to a determined temperature for a specific time (Table 1). After reaction in the autoclave, the obtained mixture was cooled down to room temperature, and a black precipitate was obtained. The product of the hydrothermal synthesis is separated magnetically from the mother liquor, followed by repeated washing with ethanol, and finally dried or re-suspended in hexane (SP, RO, and ST). In the case of the CU sample, a two-step process was followed, starting with the above described spherical particles in hexane, and exchanging the solvent with ethylacetate, as follows: 10 mL of magnetite/hexane nanoparticle suspension (10 mg/mL) was placed in a round bottom flask and 50 mL of ethyl acetate was added. This mixed solvent nanoparticle suspension was sonicated using an ultrasonic probe (Branson Sonifier 450, Danbury, CT, USA) for 10 min. Finally, the hexane solvent was evaporated under vacuum at 55 °C.

### 3.3. TEM Characterization: Morphology

The morphology of the nanoparticles was analyzed by transmission electron microscopy (TEM) using a High-Resolution LIBRA 120 Plus Carl Zeiss microscope (Oberkochen, Germany). TEM images were analyzed with J-Image software (ImageJ2 online version from University of Wisconsin, Madison, WI, USA) in order to calculate the particle size distribution of the dried NPs (*n* > 1000 nanoparticles).

### 3.4. X-ray Diffraction Measurements

The crystalline phase was identified by recording X-ray powder diffraction patterns (XRD) of the dry, washed samples using a Bruker D8 Advance diffractometer (Berlin, Germany) equipped with a CuKα radiation source (λ = 1.5406 Å) and a Bruker LINXEYE detector. Analysis was carried out at 25 °C, 40 kV and 40 mA. The 2θ measured range was 20°–70°, at 0.02° steps, with a measurement time of 576 s/step.

### 3.5. Electrophoretic Mobility: Isoelectric Point Determination

Electrophoretic mobility measurements were carried out in a Zetasizer Nano-ZS (Malvern Instruments, Worcestershire, U.K.) at 25 °C. Samples were prepared by simply adding 0.5 mL of MNPs suspensions (10 mg/mL) in TMAH 25%wt solution to 50 mL 5 mM KNO_3_ water solution (fixed ionic strength) until finally obtaining a slightly turbid solution adequate for this type of determination. The pH of the suspensions was then adjusted by adding a suitable amount of KOH (0.01 or 0.1 M) or HNO_3_ (0.01 or 0.1 M). For each suspension, 5 measuring runs were taken, with 11 cycles in each run.

### 3.6. Magnetic Properties

Magnetization cycles were obtained at room temperature (20 °C) in an MPMS-XL SQUID magnetometer (Quantum Design, San Diego, CA, USA). Between 1 and 3 mg dried samples were used for these measurements.

### 3.7. Magnetic Hyperthermia Determinations

The alternating currents necessary to produce the magnetic fields applied to the samples were produced by a Royer-type oscillator with 8 turn coils (20 mm in diameter and 45 mm in length) made of copper tube 6 mm in diameter, refrigerated with 25 °C water. The magnetic field strength H_0_ was 16.2 kA/m (measured with a NanoScience Laboratories Ltd. Probe (UK), with 10 μT resolution) in the center of the coil, where samples were located. Five frequencies were used, namely 136, 146, 160, 180, and 205 kHz.

For the determination of the heat released by the particles under the action of the field, 0.5 mL suspensions containing 10 mg/mL of each sample were placed in Eppendorf tubes, thermally isolated, and located in the center of the coil. Temperature *T* changes vs. time were registered at 1 Hz sampling rate with an optical fiber thermometer (Optocon AG, Dresden, Germany) connected to a computer. The experiments were always performed in samples pre-thermostated at 25 °C.

The heating efficiency (as evaluated by the specific absorption rate, SAR) of the nanoparticles was calculated from the initial slope *dT/dt* (first 30 s after the magnetic field was switched on), of the temperature vs. time data using the following equation [25,36]:(1)$$SAR=\frac{CV_{s}}{m}\frac{dT}{dt}$$
where *C* is the volume specific heat capacity of the sample ($C_{H_{2}O}$ = 4185 J/LK), *V_s_* = 0.5 mL is the sample volume, and *m* is the mass of magnetic material (5 mg in the present investigation). As the hyperthermia response is also dependent on the frequency f and strength H_0_ of the magnetic field, hence on the particular measuring system used, it is sometimes preferred to use an intensive, device-independent quantity to describe the hyperthermia efficiency for a given sample in terms of the so-called *intrinsic loss power* or *ILP* [31], typically expressed in [nHm^2^kg^−1^] and given by:(2)$$ILP=\frac{SAR}{fH_{0}^{2}}$$

## 4. Conclusions

Four different morphologies of magnetite nanoparticles have been synthesized using similar methodologies, obtaining comparable particle sizes. The reaction variables such as solvents used, temperature, and duration of reactions influence the shape of the nanoparticles. The synthesized nanoparticles showed an optimal particle size to be used as hyperthermia agents. The heating capacities of these superparamagnetic nanoparticles have been characterized and investigated in terms of SAR/ILP. A modulation of the saturation of magnetization and hyperthermia response were obtained in function of the morphology of the nanoparticles used. Cuboidal nanoparticles exhibited higher heating absorption properties and intrinsic loss power than nanospheres or nanorods, and the highest ILP described in the literature is for nanocubes with the same magnetic material at these low frequencies. It is suggested that this morphology could be optimum for hyperthermia using relatively low magnetic field strength and frequency, which is clinically preferred. These magnetic quasicubic nanoparticles can be a very powerful tool in biomedical applications.

## Figures and Tables

**Figure 1 nanomaterials-11-00627-f001:**
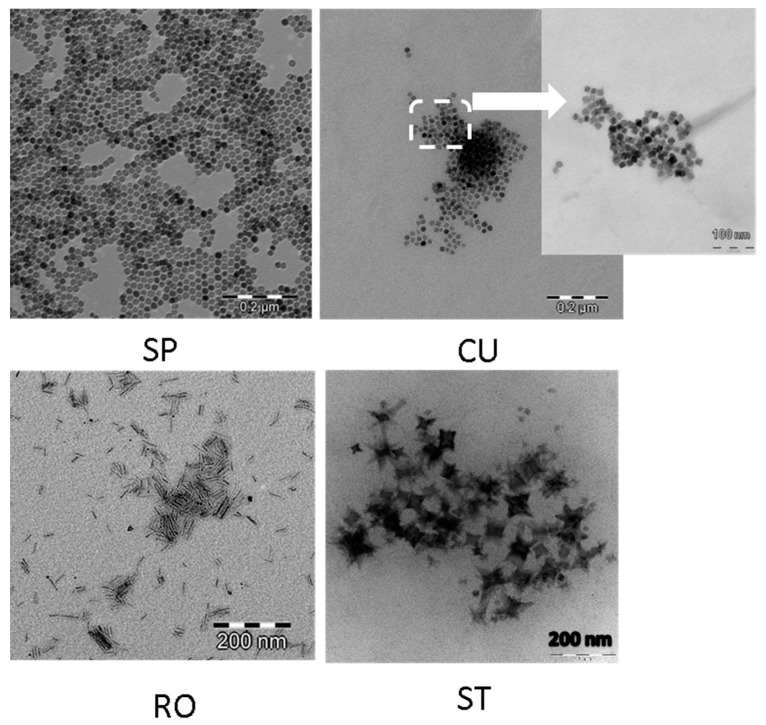
TEM images of magnetite nanoparticles obtained by hydrothermal method. SP: spheres; CU: cuboidal; RO: rods; ST: stars.

**Figure 2 nanomaterials-11-00627-f002:**
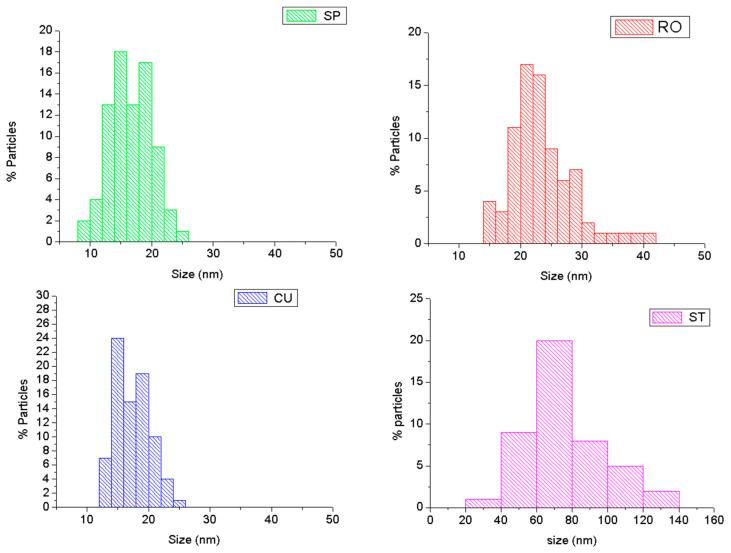
Particle size histograms of the synthesized magnetic nanoparticles (MNPs) calculated from TEM images. The histogram for the rod-like nanoparticles describes the lengths distribution (the corresponding width was 3.6 ± 0.5 nm). SP: spheres; CU: cuboidal; RO: rods; ST: stars.

**Figure 3 nanomaterials-11-00627-f003:**
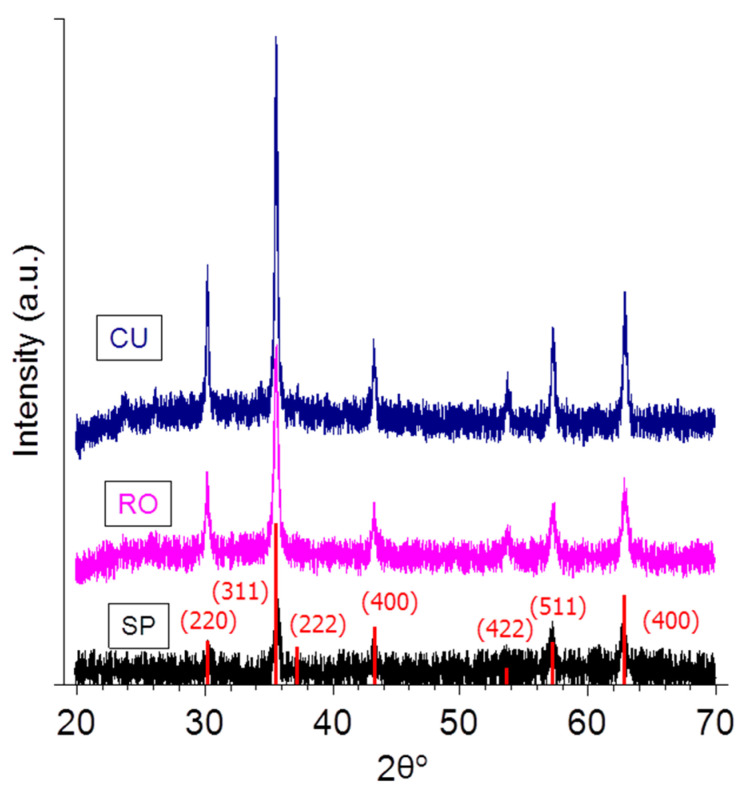
X-ray diffraction patterns of the nanoparticles. The vertical lines are the magnetite pattern (RRuff data base, ID R06111157).

**Figure 4 nanomaterials-11-00627-f004:**
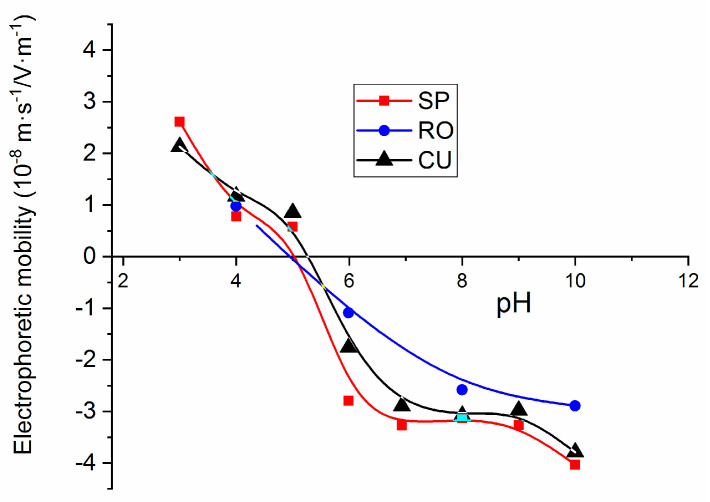
Electrophoretic mobility measurements as a function of pH for magnetite nanoparticles in TMAH 25%wt aqueous solutions.

**Figure 5 nanomaterials-11-00627-f005:**
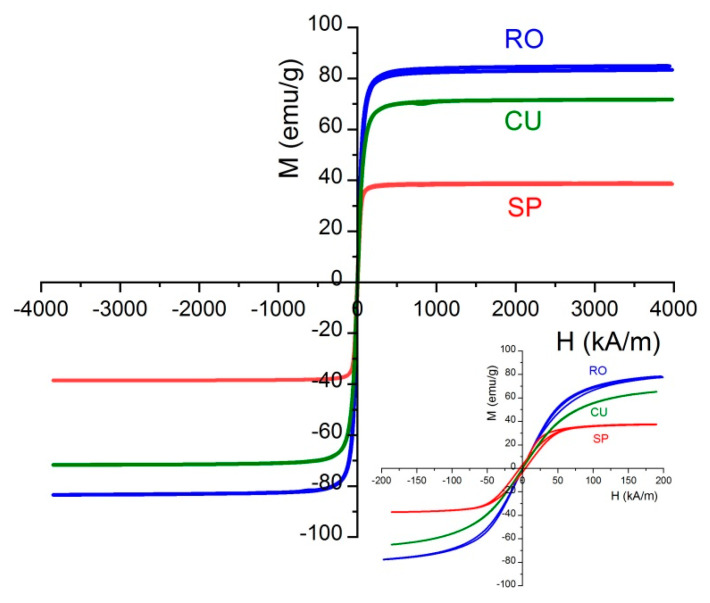
Magnetization curves at 25 °C of the nanoparticle systems investigated. Inset: Low-field detail.

**Figure 6 nanomaterials-11-00627-f006:**
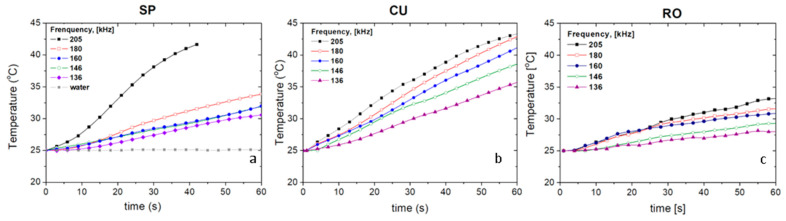
Hyperthermia response tests: temperature-time curves of samples (**a**) SP, (**b**) CU, (**c**) RO at different field frequencies and 16.2 kA/m field strength. Sample concentration was 10 mg/mL for all the nanoparticle systems.

**Figure 7 nanomaterials-11-00627-f007:**
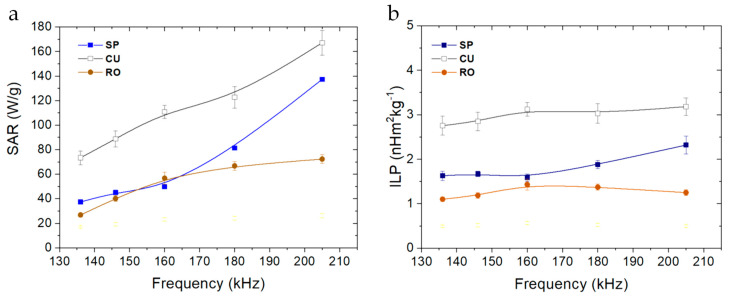
(**a**) Specific absorption rate (SAR) and (**b**) ILP values vs. frequency for the magnetite samples. Field strength: 16.2 kA/m.

**Table 1 nanomaterials-11-00627-t001:** Reaction conditions used in the preparation of magnetite nanoparticles with different morphologies. (HDA: hexadecylamine; OA: oleic acid; SP: spheres; CU: cubes; RO: rods; ST: stars). CU synthesis was carried out in two steps.

Sample	HDA (g)	OA (mL)	Fe(CO)_5_ (mL)	Solvent	T (°C)	t Reaction (h)	Morphology
SP	1.2	8	8	1-octanol (32 mL)	300	6	Spheres
CU	1.2	8	8	1st Step: 1-octanol (32 mL) 2nd Step: re-suspended in ethyl acetate (50 mL)	300	6	Cuboidals
					Room temperature		
RO	1.2	8	8	1-octanol (32 mL)	150 & 300	2 h & 4 h	Rods
ST	0.6	8	4	Benzyl ether (16 mL)	300	12	Stars

## Data Availability

The data is available on reasonable request from the corresponding author.

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
