# Peer review of "Modulation of the Magnetic Hyperthermia Response Using Different Superparamagnetic Iron Oxide Nanoparticle Morphologies"

_nanomaterials, 2021, doi:10.3390/nano11030627_

Round 1
Reviewer 1 Report
Dear Editor,
The authors present their studies on the influence of particles' morphology on magnetic hyperthermia of magnetite. The introduction is very well and detailed described the criteria that must meet the materials for the application of magnetic hyperthermia in medicine.
The synthesis conditions of the individual samples must be presented more clearly in subsection 3.2. It is not clear what were the difference in the synthesis condition for obtaining samples with spherical and cubic particle shape. In Table 1 were shown that the temperature and reaction time were the same for these two samples.
Please, use the same magnification for TEM images.
Please, show the magnetization cycles at low magnetic field.
Best regards
Author Response
COMMENT 1:
The authors present their studies on the influence of particles' morphology on magnetic hyperthermia of magnetite. The introduction is very well and detailed described the criteria that must meet the materials for the application of magnetic hyperthermia in medicine.
The synthesis conditions of the individual samples must be presented more clearly in subsection 3.2. It is not clear what were the difference in the synthesis condition for obtaining samples with spherical and cubic particle shape. In Table 1 were shown that the temperature and reaction time were the same for these two samples.
ANSWER
Thank you for your positive comments. Subsection 3.2 has been modified in order to clarify better the synthesis conditions. Cubic particle shapes were obtained in a two-step synthesis, whereby the solvent in which spherical nanoparticles are originally dispersed (hexane) is exchanged to ethyl acetate following this procedure: 10 ml of magnetite/hexane nanoparticle suspension (10 mg/ml) were placed in a round bottom flask and 50 ml of ethyl acetate were added. This mixed solvent nanoparticle suspension was sonicated using an ultrasonic probe (Branson Sonifier 450) during 10 minutes. Finally, the hexane solvent was evaporated under vacuum at 55 ºC. This procedure has been added to the revised version of the manuscript (page 13).
COMMENT 2:
Please, use the same magnification for TEM images.
ANSWER
According with the reviewer comment, Figure 1 has been modified by adding images of the four nanoparticle morphologies with the same TEM magnification.
COMMENT 3:
Please, show the magnetization cycles at low magnetic field.
ANSWER
Low-magnetic field details are included as insets in Fig. 5. Comments on these new data appear in page 9 of the revised manuscript. They are reproduced below for the reviewer’s convenience:
The low-field details included show that hysteresis is negligible, and only spheres show a measurable coercivity, below 5 kA/m. The same can be said about remanence: the particles have zero remanent magnetization, except again spheres, also below 5 kA/m. It can be said hence that the particles display superparamagnetic behavior.

Reviewer 2 Report
Title: Modulation of the magnetic hyperthermia response using different magnetite nanoparticle morphologies
In this work, the authors describe a synthesis method that allows obtaining MNPs with different morphologies, investigating four different morphologies (spheres, cubes, nanorods and stars) as concern the magnetic hyperthermia response, by applying a magnetic field of 20 kA/m amplitude and frequency in the range 136-205 kHz. Cubic MNPs displayed the maximum SAR with respect to the other ones.
Some editing by a native English speaker may help clear up some areas. The work is well structured and the proposed goals were achieved. The methods described comprehensively. The interpretations and conclusions justified by the results. Moreover, the manuscript should be improved as recommended below. In order to improve this article, the authors should put their attention on some issues.
In the introduction the Authors cite some articles about the coadiuvant role of hyperthermia with other therapies.
I suggest citing:
- Xie, Liqin, et al. “Construction of small-sized superparamagnetic Janus nanoparticles and their application in cancer combined chemotherapy and magnetic hyperthermia.” Biomaterials science, 2020, 8.5: 1431-1441.
- Brero, Francesca, et al. "Hadron therapy, magnetic nanoparticles and hyperthermia: A promising combined tool for pancreatic cancer treatment." Nanomaterials10 (2020): 1919.
- Lu, Yao, et al. “Combining magnetic particle imaging and magnetic fluid hyperthermia for localized and image-guided treatment.” International Journal of Hyperthermia, 2020, 37.3: 141-154.
Results and Discussion
Morphology and particle size distribution
Fig.1: Please, report in the caption also the meaning of the abbreviations (SP, CU, RO, ST)
Fig.2: Please, report in the caption also the meaning of the abbreviations (SP, CU, RO, ST)
Magnetization data
Please, also report the coercitive field values in the text.
Fig. 5: Please, report the temperature of the measurements, and add an inset with low fields region.
Hyperthermia response
Please, also report in the text the concentration for hyperthermia response evaluation when you report the temperature increment.
Have you checked, when you use the ILP parameter, to be in the Linear Response Theory, in which the SAR is proportional to H2?
Different minor typo corrections that should be performed.
Author Response
COMMENT 1:
In this work, the authors describe a synthesis method that allows obtaining MNPs with different morphologies, investigating four different morphologies (spheres, cubes, nanorods and stars) as concern the magnetic hyperthermia response, by applying a magnetic field of 20 kA/m amplitude and frequency in the range 136-205 kHz. Cubic MNPs displayed the maximum SAR with respect to the other ones.
Some editing by a native English speaker may help clear up some areas. The work is well structured and the proposed goals were achieved. The methods described comprehensively. The interpretations and conclusions justified by the results. Moreover, the manuscript should be improved as recommended below. In order to improve this article, the authors should put their attention on some issues.
In the introduction the Authors cite some articles about the coadiuvant role of hyperthermia with other therapies.
I suggest citing:
- Xie, Liqin, et al. “Construction of small-sized superparamagnetic Janus nanoparticles and their application in cancer combined chemotherapy and magnetic hyperthermia.” Biomaterials science,2020, 8.5: 1431-1441.
- Brero, Francesca, et al. "Hadron therapy, magnetic nanoparticles and hyperthermia: A promising combined tool for pancreatic cancer treatment." Nanomaterials10 (2020): 1919.
- Lu, Yao, et al. “Combining magnetic particle imaging and magnetic fluid hyperthermia for localized and image-guided treatment.” International Journal of Hyperthermia, 2020, 37.3: 141-154.
ANSWER:
Thank you for your positive comments. These references have been cited in the revised version. New references 11-13 have been added correspondingly.
COMMENT 2:
Results and Discussion
Morphology and particle size distribution
Fig.1: Please, report in the caption also the meaning of the abbreviations (SP, CU, RO, ST)
Fig.2: Please, report in the caption also the meaning of the abbreviations (SP, CU, RO, ST)
ANSWER:
The meaning of the abbreviations has been included in both figure captions.
COMMENT 3:
Magnetization data
Please, also report the coercitive field values in the text.
Fig. 5: Please, report the temperature of the measurements, and add an inset with low fields region.
ANSWER:
Done. The following comment was added regarding the low-field data:
The low-field details included show that hysteresis is negligible, and only spheres show a measurable coercivity, below 5 kA/m. The same can be said about remanence: the particles have zero remanent magnetization, except again spheres, also below 5 kA/m. It can be said hence that the particles display superparamagnetic behavior.
COMMENT 4:
Hyperthermia response
Please, also report in the text the concentration for hyperthermia response evaluation when you report the temperature increment.
ANSWER:
Done. As indicated on page 11 of the new version, the concentration of the samples used in hyperthermia was 10 mg/ml.
COMMENT 5:
Have you checked, when you use the ILP parameter, to be in the Linear Response Theory, in which the SAR is proportional to H2?
ANSWER:
This is a very significant question. We did not check it for all samples and all frequencies. Below is an example obtained for one of the rod-like samples by plotting the SAR as a function of the product . This was not performed systematically for all samples, but the results below and the constancy of ILP when plotted as a function of frequency is a good indication for using ILP as a representative quantity, with confidence. A sentence was added on pages 12,13 of the new version:
Previously, it was checked for some samples that the SAR was linearly dependent on the product , as expected from the linear response theory. Note how ILP is indeed roughly independent of frequency in Fig. 7, suggesting that the linear behaviour is satisfied.

Round 2
Reviewer 2 Report
All my previous comments have been carefully addressed by the authors, and the manuscript has been properly modified accordingly.
The reviewer doesn't have additional comments.
Thanks to the authors for working on this revision.
Author Response
Thank you very much for your help and support to improve the quality of my manuscript.